# Diagnosis of Uric Acid-Based Urine Sedimentation in the Bladder Using Dual-Energy CT

**DOI:** 10.3390/diagnostics13030542

**Published:** 2023-02-02

**Authors:** Christian Booz, Ibrahim Yel, Julian L. Wichmann, Simon S. Martin, Vitali Koch, Leon D. Gruenewald, Leona S. Alizadeh, Thomas J. Vogl, Tommaso D’Angelo

**Affiliations:** 1Department of Diagnostic and Interventional Radiology, University Hospital Frankfurt, 60590 Frankfurt, Germany; 2Division of Experimental Imaging, University Hospital Frankfurt, 60590 Frankfurt, Germany; 3Diagnostic and Interventional Radiology Unit, BIOMORF Department, University Hospital Messina, 98158 Messina, Italy; 4Department of Radiology and Nuclear Medicine, Erasmus MC, 3015 GD Rotterdam, The Netherlands

**Keywords:** dual energy CT, urine sedimenation, uric acid

## Abstract

Urine sedimentation in the bladder can occur in various circumstances and can lead to urinary obstruction/stasis with associated pain. It is usually diagnosed with an ultrasound; however, CT is also used to assess the amount and to further check for urinary stones. Depending on the composition, urine sedimentation and stones can be treated medically by alkalinisation of the urine with potassium sodium hydrogen citrate in the case of uric acid-based sedimentation/stones. Due to technical developments and improved material differentiation and characterisation in CT imaging, dual-energy CT allows for differentiation of uric acid from calcium, which can be used for sedimentation/stone composition analysis. Subsequently, treatment decisions can be made based on the findings in dual-energy CT.

**Figure 1 diagnostics-13-00542-f001:**
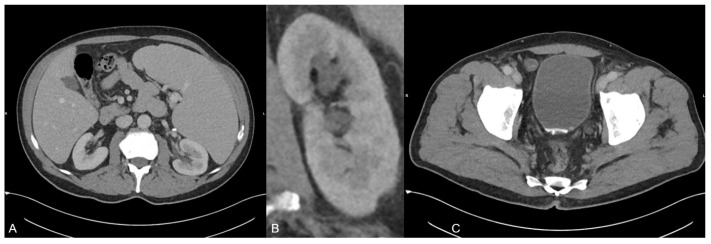
A 70-year old man was presented to our emergeny department with acute left-sided strong flank pain and clearing of fragments since some weeks ago. The only known diseases included polycythaemia vera and actinic keratosis. The polycythaemia vera was treated with aspirin and ruxolitinib (janus kinase inhibitor). The initial body check showed no signs of hematoma or inflammation. The abdomen was soft during palpation, and the pain was not influencable during the check. Laboratory blood values showed no signs of infection or ischemia. The urinary pH was 5.1. Besides the urinary pH, all other urinary tests/values (no microhematuria and no elevated leucocyte levels) were inconspicuous. The initially performed ultrasound examination revealed no presence of any acute bowel pathology including appendicitis or diverticulitis or any presence of an urinary stone. Due to the persistent strong pain, a monophasic CT scan was performed in the portal venous phase on a third-generation dual-source dual-energy CT scanner (SOMATOM Force, Siemens Healthineers, Forchheim, Germany) to check for any missed pathologies. The CT scan protocol represents a standard protocol at our institution consisting of a scan 70 s after contrast material injection (Iomeron 400 mg, injected at a dose of 1.3 mL per kilogram body weight and at a flow rate of 2 mL/s through a superficial vein of the forearm) using a dual-energy mode with tube voltages of 90 and 150 kV due to the known several advantages of dual-energy CT in clinical routine [1,2,3,4,5,6,7,8,9]. The scan revealed a known splenomegaly but no signs of a splenic rupture or infarction (**A**). In addition, no free fluid was present. Instead, a second grade urinary stasis in the left kidney (**B**) and hyperdense material in the bladder adjacent to the posterior wall in terms of an urine sedimentation (**C**) were noted. Although there was no sign of an ureter obstruction by the sediment, the sediment was the only finding which may have caused decreased flow and corresponding stasis in the CT scan. No other acute pathologies were present in this scan.

**Figure 2 diagnostics-13-00542-f002:**
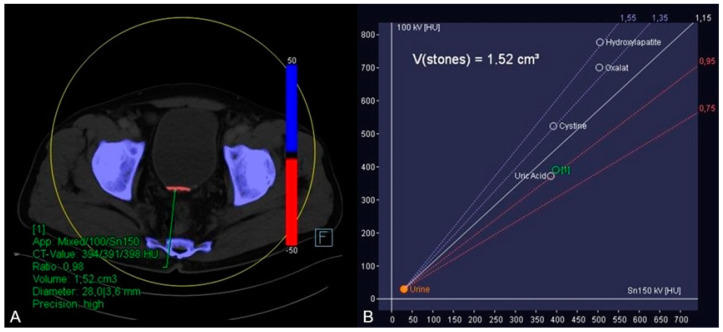
In order to further analyze the urine sedimentation, dual-energy CT-based material decomposition originally developed for kidney stone analysis differentiating uric acid from calcium was applied [10,11]. In this context, soft tissue kernel reconstructions of the 90 and 150 kV scan were fused, and dedicated postprocessing software (syngo.via, version VB10B, Siemens Healthineers) was used to generate dedicated coloured reconstructions using default settings. By application of this algorithm, it was possible to identify the red coloured uric acid in the hyperdense material (blue colour shows calcium) (**A**) by placing a circular region of interest in the hyperdense material. Furthermore, the software displayed the dual-energy ratio of the region of interest in a diagram indicating its uric acid content (**B**). Subsequently, the urologists stared a therapy including alkalinisation of the urine with potassium sodium hydrogen citrate based on the results of this dual-energy CT-based material analysis.

**Figure 3 diagnostics-13-00542-f003:**
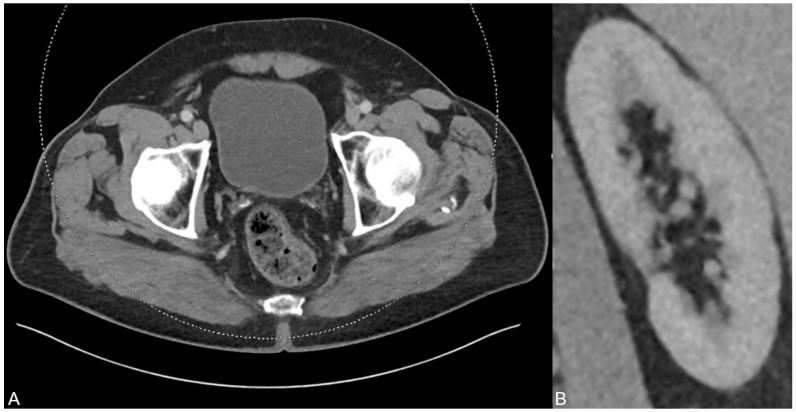
After six months and permanent therapy with potassium sodium hydrogen citrate, a control CT scan was performed showing no residues of the uric acid-based urine sedimentation in the bladder (**A**). In addition, no urinary stasis was present anymore (**B**), indicating the treatment success based on the dual-energy CT findings and analysis results.

## Data Availability

Data availability is possible upon request by C.B.

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
