# Peer review of "Diagnosis of Uric Acid-Based Urine Sedimentation in the Bladder Using Dual-Energy CT"

_diagnostics, 2023, doi:10.3390/diagnostics13030542_

Round 1

Reviewer 1 Report

The paper treats about the diagnostic role of dual-energy CT in indentification of uric acid respect to calcium stones.

If methods regarding the techincal setting of the CT are well described, the clinical case not. 

In fact in the text are not reported data about urinary pH before the start of urine alkalinization and if the Patients cleared any fragments.

Author Response

We would like to thank the reviewer for the suggestion to improve our manuscript. We hope our modifications meet your expectations.

Reviewer 2 Report

This is a very nice case study on using dual-energy CT for identifying uric acid sediment in the bladder.

1. Line 23, represents is misspelled.

2. Line 28: splenomegaly perhaps is meant here; I do not know what splenogemalia is.

3. The settling of the sediment in Figure 1C does not suggest obstruction of the ureter. Is there room in the paper to speculate on where the obstruction might have been?

Author Response

We would like to thank the reviewer for the feedback and comments to improve our manuscript. We hope the modifications/changes meet your expectations.
